# Multivessel versus Culprit-Only Percutaneous Coronary Intervention in Patients with Non-ST-Elevation Acute Coronary Syndrome

**DOI:** 10.3390/jcm11206144

**Published:** 2022-10-18

**Authors:** Tobias F. S. Pustjens, Marijke J. C. Timmermans, Saman Rasoul, Arnoud W. J. van ‘t Hof

**Affiliations:** 1Department of Cardiology, Zuyderland Medical Centre, 6419 PC Heerlen, The Netherlands; 2Department of Cardiovascular Research Institute Maastricht (CARIM), Maastricht University, 6229 ER Maastricht, The Netherlands; 3Netherlands Heart Registration, 3511 EP Utrecht, The Netherlands; 4Department of Cardiology, Maastricht University Medical Centre, 6229 HX Maastricht, The Netherlands

**Keywords:** multivessel disease, NSTE-ACS, myocardial infarction, PCI

## Abstract

Background: There is uncertainty whether multivessel (MV-PCI) or culprit-only percutaneous coronary intervention (CO-PCI) should be the treatment of choice in patients with non-ST segment elevation acute coronary syndrome (NSTE-ACS) and multivessel disease (MVD). Aims: To evaluate clinical characteristics and outcomes in these patients undergoing MV-PCI or CO-PCI at the index procedure. Methods: Data were retrieved from the nationwide Netherlands Heart Registration. All NSTE-ACS patients with MVD undergoing PCI between 1 January 2017 and 1 October 2019 were grouped into a MV-PCI or CO-PCI group. The primary endpoint was all-cause mortality at long-term follow-up (median 756 days (593–996)). Secondary endpoints were reinterventions, urgent CABG, myocardial infarction (MI) < 30 days, target vessel revascularisation (TVR) and mortality at 1 year. Propensity score matching analyses were performed. Results: In total, 10,507 NSTE-ACS patients with MVD were included into the MV-PCI (*N* = 4235) and CO-PCI group (*N* = 6272). Analysing crude data, mortality rates at long-term follow-up (10.7% vs. 10.2%; *p* = 0.383), mortality at 1 year (6.0% vs. 5.6%; *p* = 0.412) and MI <30 days (0.8% vs. 0.9%; *p* = 0.513) were similar between both groups. Reinterventions (11.1% vs. 20.0%; *p* < 0.001), urgent CABG (0.1% vs. 0.4%; *p* = 0.001) and TVR (5.2% vs. 6.7%; *p* = 0.003) occurred less often in the MV-PCI group. Survival analysis after multiple imputation and propensity score matching showed similar mortality rates at long-term follow-up (log-rank *p* = 0.289), but a significant reduction for reinterventions in the MV-PCI group (log-rank *p* < 0.001). Conclusion: NSTE-ACS patients with MVD undergoing MV-PCI have similar mortality rates at long-term follow-up compared to CO-PCI. However, improved event-free survival in terms of fewer coronary reinterventions was observed.

## 1. Introduction

Patients presenting with non-ST-elevation acute coronary syndromes (NSTE-ACS) and multivessel disease (MVD) have worse prognosis compared to those with single-vessel obstructive coronary artery disease [1]. Approximately half of the NSTE-ACS patients have MVD, which leads to the dilemma of what revascularisation strategy should be chosen (complete revascularisation by multivessel (MV-PCI) or culprit-only percutaneous coronary intervention (CO-PCI) [2]. The guidelines also cannot help as sufficient evidence from clinical trials is lacking. As such, both the American and the European guidelines do not clearly recommend which coronary revascularisation strategy should be considered for these patients [2,3].

In patients with ST-elevation acute coronary syndrome (STE-ACS) and MVD, it was shown that MV-PCI compared to CO-PCI resulted in reduced major adverse cardiovascular events, mainly driven by reduced repeated revascularisation [4,5,6,7,8]. The COMPLETE trial was the first to demonstrate that complete revascularisation resulted in improved clinical outcome (death or myocardial infarction [MI]) compared to CO-PCI [9]. Whether this can be translated to NSTE-ACS patients, remains unclear. Previous studies showed that incomplete revascularisation in NSTE-ACS with MVD was associated with worse short- and long-term prognosis, while others report no significant differences in long-term outcomes between the groups [10,11,12,13,14,15].

Since findings of previous cohort studies have been inconsistent, we aimed to evaluate clinical characteristics as well as short- and long-term outcomes in patients with NSTE-ACS and MVD undergoing MV-PCI or CO-PCI at the index procedure in a large nationwide registry.

## 2. Materials and Methods

### 2.1. Source of Study Data

The Netherlands Heart Registration (NHR) is a nationwide quality registry in which baseline, procedural and outcome data across all invasive cardiac interventional, electrophysiological and surgical procedures are registered. Details on the process of data acquisition, completeness, data quality and analysis of the NHR were described previously [16]. In brief: the aim of the NHR is to evaluate current practices in the treatment of heart disease, through all stages of the treatment process: from diagnosis to many years after the intervention. Data are validated using multiple methods. For example, hospitals receive an automated data quality report directly after upload of the data and each year a monitor visit (audit) is conducted to compare the data submitted to the NHR with the information in the medical records. In this nationwide registry, only a limited number of demographic and procedure-related variables are collected for feasibility reasons. The use of data in the NHR database for research purposes has been approved by the Medical Research Ethics Committees United (reference number W19.270) that issued a waiver for informed consent for the current analysis of anonymised data.

### 2.2. Study Population

All patients ≥ 18 years old presenting with NSTE-ACS and MVD (defined as coronary stenosis ≥ 70% in ≥ 2 major coronary arteries or in 1 major coronary artery plus the first major side branch with a diameter of at least 1.5 mm) and in whom the type of revascularisation strategy was known, were eligible for analysis. In case more than one revascularisation was registered for a unique patient, the first revascularisation was counted as the index procedure. Exclusion criteria were presentation with an out-of-hospital cardiac arrest, cardiogenic shock at baseline, a history of previous CABG and chronic total occlusion of ≥1 native coronary artery.

Patients were grouped according to their revascularisation strategy at the index procedure (MV-PCI vs. CO-PCI); if only one main vessel was treated, the patient was included in the CO-PCI group, if ≥2 main vessels were treated the patient was included in the MV-PCI group.

### 2.3. Study Design

For this observational multicentre cohort study, information related to the patients undergoing PCI from 1 January 2017 until 1 October 2019 were extracted from 29 PCI centres in The Netherlands. Obtained patient demographics were age, gender, a history of diabetes mellitus, previous PCI, left ventricular ejection fraction and reduced renal function defined as an estimated glomerular filtration rate (eGFR) of <60 mL/min/1.73 m^2^. Furthermore, procedure-related data (treated vessels and type of treatment) and outcome measures (all-cause mortality and reintervention at long-term follow-up, all-cause mortality and target vessel revascularisation (TVR) at 1 year, urgent CABG within 1 day and recurrent myocardial infarction (MI) <30 days). TVR was defined as recurrent revascularisation of the same vessel(s) as treated during the index PCI).

The NHR has robust and near-complete data on all-cause mortality. Each participating centre obtained mortality data predominantly by consulting the regional municipal administration registry. Elective and non-elective coronary reinterventions were evaluated by linking our NSTE-ACS dataset with the full PCI dataset (including revascularisation in STE-ACS and elective patients) and the NHR cardiac surgery dataset, which covers all coronary artery bypass grafting (CABG) procedures in The Netherlands. In this way, no reinterventions were missed. To account for planned revascularisations, all elective reinterventions within the first 6 weeks after the index PCI were not counted as an event. All the above-mentioned data were collected by the centre performing the initial treatment. Follow-up data were acquired until 1 February 2021.

### 2.4. Outcomes

The primary endpoint was all-cause mortality at long-term follow-up with a maximum of 4 years’ follow-up. Secondary endpoints were all reinterventions at long-term follow-up, urgent CABG within 1 day, recurrent MI < 30 days, TVR at 1 year and 1-year mortality.

### 2.5. Statistical Analysis

Categorical variables were expressed as frequencies and percentages and were analysed by the binary logistic regression with corresponding odds ratio [OR] including the 95% confidence interval [CI]. Normality of distribution depended on the skewness and kurtosis. Continuous variables with Gaussian distributions were tested with the Student t-test and depicted with mean ±standard deviations. Baseline variables with non-Gaussian distributions were compared using the Mann–Whitney U test and summarized with medians and interquartile ranges [IQR]. Multiple imputation was used to accommodate the presence of missing values for demographic variables (except for left ventricular ejection fraction and dialysis since this data was missing in >10% and was assumed not missing at random). The dataset was imputed 5 times under the assumption of data being missing at random. Neither missing outcome data nor the grouping variable were imputed.

To prevent potential treatment bias or other confounders in both groups, a propensity score–matched analysis was carried out using a non-parsimonious logistic regression model comparing MV-PCI versus CO-PCI for each imputed dataset. The following variables were included in the model: age, gender, diabetes mellitus, creatinine clearance < 60 mL/min/1.73 m^2^, previous PCI, previous myocardial infarction, radial access, left main (LM) treatment and types of vessel treatment. After deriving a propensity score, individual patients were matched using nearest neighbour matching in a 1:1 ratio, and the calliper distance was set at 0.01. Baseline characteristics and outcome data were presented for both the overall and the pooled propensity score–matched cohort with corresponding OR or hazard ratios (HR) with the 95% CI. For the latter cohort, Rubin’s rules were applied to pool parameter estimates. Time was measured from the index admission to outcomes (all-cause mortality or reintervention). Survival curves for all-cause mortality and reinterventions were made based upon Kaplan–Meier estimates and compared using the log-rank test.

Using the overall cohort data, a Cox proportional hazard model including a set of variables with significant differences between study groups (as revealed by univariable analysis) were incorporated to estimate the contributions of individual parameters on mortality risk and reinterventions at long-term follow-up. Age, gender and the grouping variable (MV-PCI vs. CO-PCI) were kept in the model irrespective of *p*-value. The model was built by including all potentially relevant covariates based on biological plausibility that may affect mortality or reinterventions.

For exploratory purposes, subgroup analyses were conducted by introducing a subgroup by treatment interaction term into the Cox model. Corresponding interaction *p*-values were calculated.

A two-sided alpha <0.05 was considered statistically significant for all analyses. Statistical analysis was performed using SPSS (IBM SPSS Statistics, Version 26.0. Armonk, NY, USA: IBM Corp.)

## 3. Results

### 3.1. Baseline Characteristics

Between 01.01.2017 and 01.10.2019, a total of 17,706 patients presented with NSTE-ACS in all Dutch PCI centres as listed in the appendix (Appendix A). After exclusion of patients with out-of-hospital cardiac arrest (*n* = 443), cardiogenic shock (*n* = 230), previous CABG (*n* = 2839), chronic total occlusion (*n* = 756), duplicates (*n* = 530) and missing data concerning the type of coronary intervention (*n* = 2527), a total of 10,507 patients fulfilled the inclusion criteria and were included in this analysis (Figure 1). Of those, 4235 patients (40.3%) underwent MV-PCI and 6272 (59.7%) underwent CO-PCI during the index procedure.

Baseline characteristics are presented in Table 1. In the overall cohort, patients in the MV-PCI group compared to the CO-PCI group were older (69.1 ± 11.6 vs. 68.4 ± 11.4 years; *p* = 0.004) and more likely to have renal disease (28.3% vs. 26.0%; *p* = 0.012) but had a lower prevalence of previous myocardial infarction (24.1% vs. 27.5%; *p* < 0.001) and previous PCI (25.8% vs. 32.7%; *p* < 0.001).

Radial access was used in more than 85% of the patients. In the MV-PCI group, treatment of the LAD and LCX was the predominant variant of revascularisation (49.1%), followed by treatment of the LAD and RCA (26.1%). In the CO-PCI group, treatment of the LAD (40.4%) was most prevalent, followed by the RCA (30.4%) and LCX 24.9%). In the MV-PCI group, stenting (98.0% vs. 91.2%; *p* < 0.001) and other treatments (e.g., rotablator, laser, atherectomy, thrombosuction [4.1% vs. 3.2%; *p* = 0.018]) were performed more often, whereas the occurrence of balloon dilation (8.6% vs. 8.4%; *p* = 0.654) was similar in both groups.

The baseline characteristics of the study population after imputation of missing data are presented in the Appendix A.

### 3.2. Outcome in the Overall Cohort (Crude Data)

In the overall cohort, data with regard to survival status were available for 10,442 patients (99.4%) with a median follow-up duration of 756 days (593–996). All-cause mortality at this time point was 10.7% in the MV-PCI vs. 10.2% in the CO-PCI group (*p* = 0.383). All-cause mortality within 1 year of index PCI (6.0% vs. 5.6%; *p* = 0.412) and myocardial infarction within 30 days (0.8% vs. 0.9%; *p* = 0.513) were also comparable in both the MV-PCI and CO-PCI group (Table 2).

On the other hand, in the MV-PCI group, coronary reinterventions (11.1% vs. 20.0%; *p* < 0.001), urgent CABG (0.1% vs. 0.4%; *p* < 0.001) and TVR (5.2% vs. 6.7%; *p* = 0.003) occurred less often. When elective reinterventions within 3 months after the index procedure were excluded, there were still statistically fewer reinterventions observed in the MV-PCI group at long-term follow-up (10.2% vs. 16.2%; *p* < 0.001 [data not shown]).

The Kaplan–Meier curves of the all-cause mortality and reinterventions at long-term follow-up are represented in Figure 2A,C, respectively.

### 3.3. Outcome in the Propensity Score–Matched Cohort

After multiple imputation, propensity score matching was performed for each imputed dataset to correct for the differences in baseline characteristics. Parameters used for matching were described previously. Left ventricular eject fraction (LVEF) was not included in the model due to the presence of missing data in >10%, which were assumed missing not at random. A pooled total of 7390 cases were matched in a 1:1 ratio. After matching, there were no statistically significant differences between the treatment groups (Table 1).

In the matched groups, similar findings on outcome parameters between the MV-PCI and CO-PCI group were found as observed in the overall cohort (Table 2). For all-cause mortality at long-term follow-up, the median follow-up duration was 756 days (593–996), with a mortality rate of 9.7% vs. 11.0% in the MV-PCI and CO-PCI group respectively (*p* = 0.289). Reinterventions occurred less often in the MV-PCI group (10.6% vs. 18.1%; *p* < 0.001). The Kaplan–Meier curves of all-cause mortality and reinterventions for the propensity score–matched group are depicted in Figure 2B,D. Furthermore, urgent CABG (0.1% vs. 0.2%; *p* = 0.107), TVR at 1 year (4.7% vs. 5.4%; *p* = 0.156) and all-cause mortality at 1 year (5.4% vs. 5.9%; *p* = 0.439) were equally distributed across both groups.

### 3.4. Predictors of All-Cause Mortality and Reinterventions

After multivariable regression analysis, MV-PCI remained a non-significant contributor for all-cause mortality at long-term follow-up (HR 0.91 (0.80–1.05)) (Table 3). Female gender and radial approach for coronary angiography were independent predictors for reduced all-cause mortality, whereas higher age, worse renal function, diabetes mellitus, dialysis, previous MI, LM PCI and other vessel treatment was associated with higher all-cause mortality. 

For reinterventions, MV-PCI remained an independent predictor (HR 0.56 (0.50–0.63)) for a significant reduction of reinterventions after multivariable regression analysis (Appendix A).

### 3.5. Subgroup Analysis on All-Cause Mortality and Reinterventions

Figure 3 shows the effect of MV-PCI vs. CO-PCI on the occurrence of all-cause mortality at long-term follow-up with corresponding adjusted hazards and interaction *p*-values across several subgroups including age groups, gender, diabetes mellitus, renal function, LVEF, previous PCI, previous MI and LM PCI. Females (adjusted HR 0.78 (0.61–0.99)) and non-diabetics (adjusted HR 0.82 (0.68–0.98)) tended to benefit more from a MV-PCI approach without evidence of a statistically significant interaction effect (interaction *p* = 0.112 and *p* = 0.106, respectively).

In addition, less reinterventions in those undergoing MV-PCI were consistently observed among various subgroups (Appendix A). There was a significant interaction effect observed for previous PCI status or undergoing LM PCI on reinterventions. Those without previous PCI (adjusted HR 0.49 (0.42–0.56), interaction *p* < 0.001) or without undergoing LM PCI (adjusted HR 0.51 (0.45–0.58), interaction *p* < 0.001) had more benefit from MV-PCI at the index procedure as compared to CO-PCI.

## 4. Discussion

The present prospective cohort study concerns the analysis of over 10,000 patients with NSTE-ACS undergoing PCI between 2017 and 2019. Patient characteristics and outcomes were compared between MV-PCI and CO-PCI. MV-PCI was performed in only 40% of the overall cohort, but in general they had a higher premorbid risk compared to the CO-PCI group. Mortality rates did not differ in both groups; however, patients undergoing MV-PCI were shown to have fewer coronary reinterventions at long-term follow-up, urgent CABGs within 1 day and TVR within 1 year. To exclude potential operator’s bias for the choice of revascularisation type, propensity score matching and multivariable regression analysis were performed to account for potential confounders. The main findings on clinical outcomes remained unchanged. Our findings suggest that MV-PCI at the index procedure may be considered to prevent unplanned or non-elective reinterventions.

In our study, similar mortality rates between both the MV-PCI and CO-PCI group were observed. Conflicting data exists on the benefits on long-term mortality of MV-PCI in NSTE-ACS. Whilst there were several studies showing no mortality differences in MV-PCI, others were able to demonstrate a significant reduced mortality risk [15,17,18,19,20,21,22]. A possible explanation could be that patients undergoing incomplete revascularisation are more likely to have a greater burden of comorbidities that may also impact the completeness of revascularisation. In observational studies, it is challenging to account for all underlying reasons why an operator might choose a revascularisation strategy. In our study, subgroup analysis for the mortality endpoint revealed in females or non-diabetics MV-PCI may have beneficial effects on mortality. However, interaction *p*-values were not significant and since the subgroup analysis were performed for exploratory purposes, they should be interpreted with caution.

Rathod et al. (2018) observed a significant reduction of mortality risk at long-term follow-up in single-stage MV-PCI [15]. However, this mortality benefit was present beyond 6 months of index PCI possibly due to an increased in-hospital mortality for these patients. Since our study had a shorter mean follow-up time (2.1 years in our cohort compared to 4.1 years in the study by Rathod et al.), this might have influenced the translatability of our findings on longer term mortality.

On the contrary, randomised, controlled trials in STE-ACS consistently found no reduction in mortality rates. The reduction of the primary endpoint in the Complete vs. Culprit-only Revascularisation to Treat Multivessel Disease After Early Percutaneous coronary intervention for STE-ACS (COMPLETE) trial was completely driven by a reduction of non-fatal myocardial infarctions, whilst mortality rates were equal in both groups which may suggest that complete revascularization does not contribute to an improvement in LVEF [9]. Therefore, our observation of similar mortality rates in both revascularisation groups are in line with the COMPLETE trial.

Nonetheless, we were able to demonstrate a major reduction in reinterventions in those undergoing MV-PCI. The reduced revascularisation rate persisted when all elective revascularisation within 3 months of index PCI were excluded (10.2% in the MV-PCI vs. 16.2% in the CO-PCI group; *p* < 0.001 [data not shown]). Furthermore, subgroup analysis revealed a reduction of revascularisation across almost all subgroups with a significant interaction favouring MV-PCI in those without previous PCI and without undergoing LM PCI.

This major reduction in reinterventions, without a negative impact on mortality risk, in MV-PCI might have a beneficial impact to reduce pressure on health care systems and health care costs due to the reduced need for recurrent, elective and non-elective, hospitalisations. This might also lead to improved self-reported quality-of-life and angina severity. In the future, since SF-36 questionnaires are currently part of the NHR data acquisition, quality-of-life differences across revascularisation groups could be determined. Large-scale randomised, controlled trials are warranted not only to demonstrate the impact of MV-PCI on hard clinical endpoints, but also to evaluate differences in physical and psychological health, and cost-effectiveness for different revascularisation strategies.

Whilst there is a large body of evidence present in patients with ST-elevation ACS (STE-ACS) suggesting complete revascularisation to be superior to CO-PCI [4,6,8,9,23], there is a lack of sufficient evidence supporting the benefits of complete revascularisation in NSTE-ACS. Whether the results from the aforementioned studies could be translated to NSTE-ACS remains unclear. Accordingly, the recently published AHA guidelines on coronary artery revascularisation fails to give a recommendation on completeness of revascularisation in NSTE-ACS with MVD, although the latest ESC-guidelines state that complete revascularisation in NSTE-ACS with MVD may be considered [2,3]. However, this is based on outdated studies comparing the benefit of early intervention versus a conservative approach [24,25,26]. It is unknown whether stenting of non-culprit lesions in NSTE-ACS with MVD improves outcomes since robust data confirmed by randomised, controlled studies is lacking. An important issue in NSTE-ACS with MVD is that it may be challenging to identify the culprit lesion, whilst in STE-ACS this is most obvious. The use of visual angiographic clues as well as intracoronary haemodynamic assessment by FFR or intraluminal imaging by optical coherence tomography or intravascular ultrasound, or the use of non-invasive imaging modalities such as ECG, echocardiography and cardiac magnetic resonance imaging could contribute to the identification of the culprit lesion. In 40%, multiple vulnerable plaques, sometimes affecting >1 vascular territory, are present [27]. This may explain some benefits of complete revascularisation, since in CO-PCI other potential culprit arteries may be left untreated. On the other hand, there are concerns regarding the potential risk of contrast-induced nephropathy, periprocedural myocardial infarction and the risk of stent thrombosis due to the prothrombotic state in ACS.

To date, the SMILE trial is the only randomised, controlled study in this field that concluded that single-stage MV-PCI in NSTE-ACS patients was superior to multistage MV-PCI. However, it does not address whether MV-PCI is beneficial to CO-PCI at the index procedure and what the added value of haemodynamic assessment of non-culprit lesions is [28]. The current ongoing South Limburg Myocardial Infarction (SLIM) trial will investigate the clinical effects of ischaemia-driven complete revascularisation by fractional flow reserve (FFR) compared to CO-PCI during the index procedure (clinicaltrials.gov NCT03562572) [29]. In addition, the BIOVASC study will randomise ACS patients with MVD to complete revascularisation or culprit-only plus staged complete revascularisation (clinicaltrials.gov NCT03621501) and the FIRE trial will provide evidence on whether a specific revascularisation strategy should be applied to elderly patients presenting with ACS and MVD [30,31]. These studies will give further insights into refining the treatment algorithm in these patients.

### Strengths and Limitations

The strength of this study is that it included patients from all 29 PCI centres throughout The Netherlands representing a true representation of daily practice in a diverse ethnic and socioeconomic population. Outcome data was widely available, with robust data on mortality due to reliable tracking in government administration systems.

This study has several limitations, primarily related to its observational design and may have potential bias and unmeasured confounding. First, a proportion of the study population was ineligible for further analysis since data regarding revascularisation strategy, especially in the first year of this study, was missing. Due to the ongoing process of optimisation of the national registry, data became almost complete during the course of this study. Furthermore, for feasibility reasons, only a relatively small subset of baseline and procedural characteristics are collected in this registry. Therefore, we were not able to correct for all possible confounders nor could we provide more insight in the extensiveness of coronary artery disease or performed invasive physiological diagnostic methods. This may have had influence on determining the appropriate culprit stenosis since this is more challenging as compared to the STE-ACS population.

Secondly, MVD was based on the operator’s judgment and was therefore subjective, comparable to common practice. The cut-off value of a significant stenosis of ≥70% was chosen by the PCI registration committee which consisted of experienced interventional cardiologists from the participating PCI centres. However, some studies used other cut-off values which may limit the comparability of our study to others.

Thirdly, our study does not answer the question whether complete revascularisation needs to be performed during a single-staged or a multistaged procedure, since the intended multistage procedures were not captured in this registry. Therefore, some degree of treatment bias could not be prevented since some patients attributed to the CO-PCI group may have been incompletely revascularised. To account for possible planned revascularisations (especially for those included in the CO-PCI group), all elective reinterventions within the first 6 weeks after the index PCI were excluded.

Fourth, although the mean follow-up time is limited, The Netherlands Heart Registration has robust data on outcome measures from 2017 onwards. Due to the absence of (near) complete data before this time, we were unable to present longer follow-up data.

## 5. Conclusions

NSTE-ACS patients with MVD undergoing MV-PCI have similar mortality rates at long-term follow-up compared to CO-PCI. However, improved event-free survival in terms of fewer coronary reinterventions was observed.

## Figures and Tables

**Figure 1 jcm-11-06144-f001:**
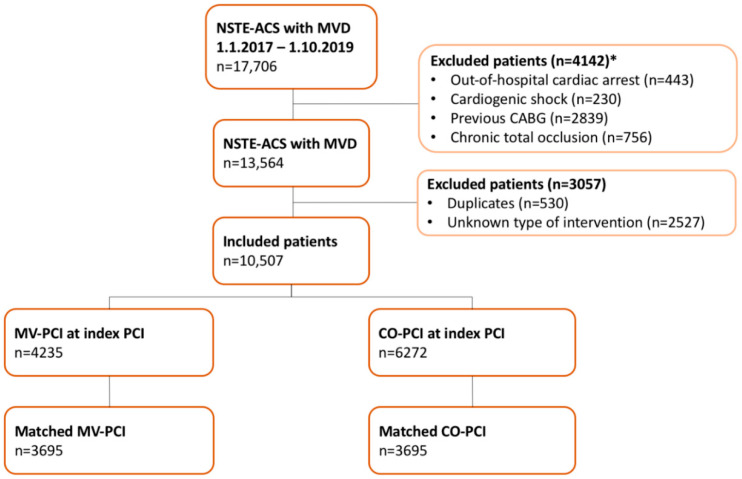
Flow-chart of included NSTE-ACS patients undergoing PCI. CABG, coronary artery bypass grafting; CO-PCI, culprit-only PCI; NSTE-ACS, non-ST-segment elevation acute coronary syndrome; MVD, multivessel disease; MV-PCI, multivessel PCI; PCI, percutaneous coronary intervention. * As some patients met multiple exclusion criteria, numbers do not sum to 4142.

**Figure 2 jcm-11-06144-f002:**
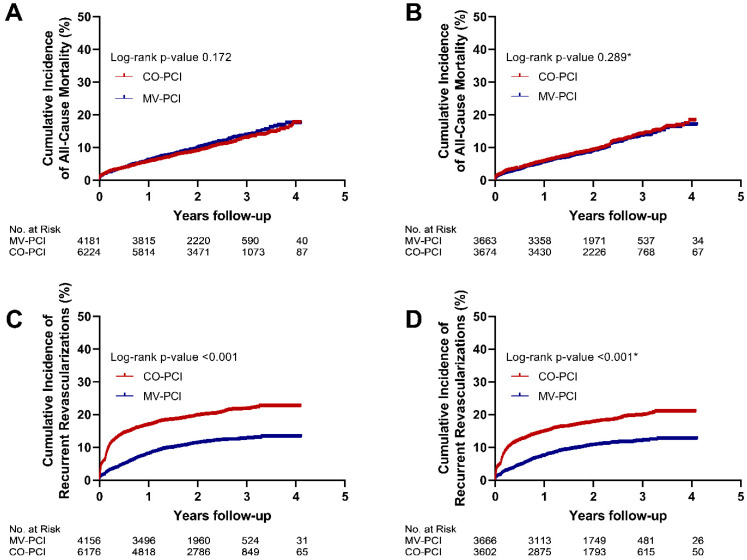
Kaplan–Meier curve showing the all-cause mortality and reinterventions at long-term follow-up in the overall cohort (**A**,**C**) and propensity score–matched cohort of multiple imputation dataset 1 (**B**,**D**). Abbreviations as in Figure 1. * Represents the pooled *p*-value for all five propensity score–matched cohorts.

**Figure 3 jcm-11-06144-f003:**
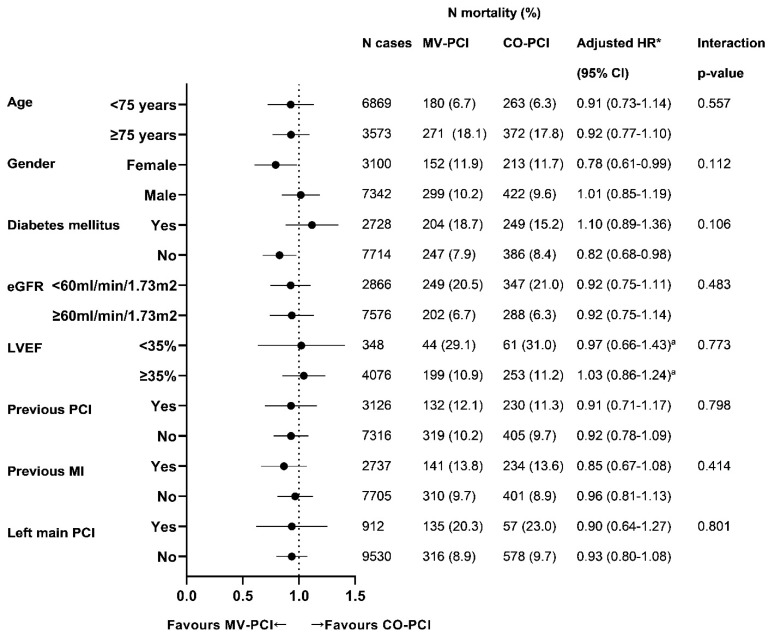
Subgroup analysis on mortality at long-term follow-up between MV-PCI and CO-PCI. Values are *n* (%), or HR (95% CI). CI, confidence interval; HR, hazard ratio; other abbreviations as in Figure 1, Table 1 and Table 3. * Covariates used for correction: age, gender, creatinine level < 60 mL/min/1.73 m^2^, diabetes mellitus, dialysis, previous MI, radial approach, left main PCI and vessel treatment. ^a^ Data depicted represent unadjusted HR due to missing data assumed to be not missing at random.

**Table 1 jcm-11-06144-t001:** Baseline characteristics.

	Overall Cohort				Propensity Score Matched Cohort			
	MV-PCI *n* = 4235	CO-PCI *n* = 6272	OR (95%-CI)	*p*-Value	MV-PCI *n* = 3695	CO-PCI *n* = 3695	OR (95%-CI)	*p*-Value
Age	69.1 ± 11.6	68.4 ± 11.4	NA	0.004	68.6 ± 11.6	68.5 ± 11.6	NA	0.839
Gender, female	1286 (30.4)	1824 (29.1)	1.06 (0.98–1.16)	0.157	1105 (29.9)	1087 (29.4)	1.02 (0.92–1.14)	0.661
eGFR < 60 mL/min/1.73 m^2^	1136 (28.3)	1579 (26.0)	0.89 (0.82–0.98)	0.012	1007 (27.2)	1010 (27.3)	1.00 (0.90–1.13)	0.940
Diabetes mellitus	1086 (25.9)	1634 (26.3)	0.98 (0.90–1.07)	0.701	937 (25.4)	963 (26.1)	0.96 (0.85–1.09)	0.545
LVEF < 35%	151 (7.6)	198 (8.0)	1.06 (0.85–1.33)	0.588	129 (7.4)	127 (8.5)	1.16 (0.90–1.52)	0.252
Dialysis	37 (1.0)	51 (0.9)	1.09 (0.71–1.67)	0.683	29 (0.9)	29 (0.9)	0.94 (0.54–1.64)	0.837
Previous MI	1008 (24.1)	1702 (27.5)	0.84 (0.77–0.92)	<0.001	882 (23.9)	889 (24.1)	0.99 (0.89–111)	0.867
Previous PCI	1084 (25.8)	2015 (32.7)	0.72 (0.66–0.78)	<0.001	934 (25.3)	919 (24.9)	1.02 (0.92–1.14)	0.687
Radial approach	3410 (85.5)	5027 (86.6)	0.91 (0.81–1.02)	0.117	3140 (85.0)	3060 (82.8)	1.15 (0.75–1.77)	0.456
Left main PCI	666 (15.7)	252 (4.0)	4.46 (3.83–5.18)	<0.001	252 (6.8)	251 (6.8)	1.01 (0.83–1.22)	0.962
Treated vessels			-*	-*			-*	-*
LAD		2537 (40.4)				2282 (61.8)		
LCX		1562 (24.9)				896 (24.2)		
RCA		1904 (30.4)				265 (7.2)		
LAD + LCX	2081 (49.1)				1693 (45.6)			
LAD + RCA	1106 (26.1)				1074 (29.1)			
LCX + RCA	630 (14.9)				607 (16.4)			
LAD + LCX + RCA	416 (9.8)				319 (8.6)			
Lesions treated	2 (2–3)	1 (1–1)	NA	<0.001	2 (2–3)	1 (1–2)	NA	<0.001
Stent treatment			4.72 (3.74–5.95)	<0.001			1.06 (0.78–1.44)	0.712
DES	3501 (84.4)	5083 (88.9)			3037 (84.1)	2978 (82.5)		
BMS	6 (0.1)	19 (0.3)			4 (0.1)	19 (0.5)		
BRS	0 (-)	6 (0.1)			0 (-)	0 (-)		
Unknown	643 (15.5)	611 (10.7)			571 (15.8)	611 (16.9)		
Balloon dilatation	365 (8.6)	525 (8.4)	1.03 (0.90–1.19)	0.654	230 (6.2)	232 (6.3)	0.90 (0.82–1.20)	0.989
Other treatment	174 (4.1)	203 (3.2)	1.28 (1.04–1.58)	0.019	80 (2.1)	84 (2.3)	0.94 (0.58–1.52)	0.789

Values are mean ± standard deviation, *n* (%) or median (interquartile range). BMS, bare metal stent; BRS, bioresorbable vascular scaffold; CI, confidence interval; DES, drug eluting stent; eGFR, estimated glomerular filtration rate; LVEF, left ventricular ejection fraction; MI, myocardial infarction; NA, not applicable; OR, odds ratio; other abbreviations as in Figure 1. * *p*-value could not be determined since there was no comparator.

**Table 2 jcm-11-06144-t002:** Outcome data.

	Overall Cohort				Propensity Score–Matched Cohort			
	MV-PCI *n* = 4235	CO-PCI *n* = 6272	OR or HR (95%-CI)	*p*-Value	MV-PCI *n* = 3696	CO-PCI *n* = 3696	OR or HR (95%-CI)	*p*-Value
Urgent CABG (<1 day)	3 (0.1)	26 (0.4)	0.17 (0.052–0.56)	0.004	3 (0.1)	9 (0.2)	0.33 (0.088–1.27)	0.107
MI ≤ 30 days	24 (0.8)	44 (0.9)	0.85 (0.51–1.40)	0.513	21 (0.8)	28 (1.0)	0.77 (0.43–1.39)	0.388
TVR at 1 year	199 (5.2)	404 (6.7)	0.77 (0.65–0.92)	0.003	151 (4.6)	189 (5.4)	0.86 (0.69–1.06)	0.156
Mortality at 1 year	253 (6.0)	351 (5.6)	1.08 (0.92–1.27)	0.351	197 (5.4)	216 (5.9)	0.92 (0.76–1.13)	0.439
Mortality at long-term follow-up	451 (10.7)	635 (10.2)	1.09 (0.96–1.23)	0.173	358 (9.7)	404 (11.0)	0.92 (0.80–1.07)	0.289
Reinterventions at long-term follow-up	470 (11.1)	1254 (20.0)	0.53 (0.47–0.59)	<0.001	389 (10.5)	668 (18.1)	0.57 (0.50–0.64)	<0.001
PCI	409 (87.0)	1035 (82.5)			340 (87.4)	585 (87.6)		
CABG	61 (13.0)	219 (17.5)			49 (12.6)	83 (12.4)		

Values are *n* (%). HR, hazard ratio; TVR, target vessel revascularisation; other abbreviations as in Figure 1 and Table 1.

**Table 3 jcm-11-06144-t003:** Cox proportional model of multivariable analysis of predictors of all-cause mortality at long-term follow-up.

	Univariable	Multivariable *
Age	1.07 (1.06–1.07)	1.05 (1.04–1.06)
Gender, female	1.21 (1.07–1.37)	0.85 (0.74–0.98)
eGFR < 60 mL/min/1.73 m2	3.55 (3.15–4.00)	2.00 (1.72–2.31)
Diabetes mellitus	2.08 (1.84–2.34)	1.73 (1.51–1.97)
LVEF	3.13 (2.53–3.87)	NA
Dialysis	7.29 (5.45–9.75)	3.18 (2.33–4.33)
Previous MI	1.57 (1.39–1.76)	1.37 (1.17–1.60)
Previous PCI	1.21 (1.06–1.37)	0.90 (0.77–1.05)
Radial approach	0.62 (0.39–0.76)	0.67 (0.57–0.78)
Left main PCI	2.48 (2.12–2.90)	1.54 (1.28–1.84)
Vessel treatment		
Stent treatment	0.81 (0.65–1.01)	NA
Balloon dilatation	1.30 (1.07–1.57)	NA
Other treatment	1.61 (1.24–2.10)	1.47 (1.10–1.95)
Multivessel PCI	1.09 (0.96–1.23)	0.91 (0.80–1.05)

Values are HR (95% CI). eGFR, estimated glomerular filtration rate; LVEF, left ventricular ejection fraction; other abbreviations as in Figure 1 and Table 2. * Covariates used for correction: age, gender, creatinine level < 60 mL/min/1.73 m^2^, diabetes mellitus, dialysis, previous MI, previous PCI, radial approach, left main PCI, other treatment and multivessel PCI.

## Data Availability

Restrictions apply to the availability of these data. Data were obtained from The Netherlands Heart Registration and are available upon request.

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
