# Peer review of "Multivessel versus Culprit-Only Percutaneous Coronary Intervention in Patients with Non-ST-Elevation Acute Coronary Syndrome"

_jcm, 2022, doi:10.3390/jcm11206144_

Round 1

Reviewer 1 Report

I am very interested in this study.
This study is a significant clinical trial to determine the PCI strategy for NSTEMI. I mostly agree with you regarding the study methodology and results in this study.

I had two concerns that I would like you to address.

1. I understand that PCI performed within 6 weeks of index PCI was excluded from the event due to the possibility of elective staged PCI. Also, in line 209 (p. 7), it is stated that the frequency of reintervention was higher in CO-PCI, even if reintervention within 3 months is excluded. However, looking at the Kaplan-Meier curve, it appears that reintervention within 3 months forms the difference between the two groups. (The text states that there is a 6% difference after 3 months, but the Kaplan-Meier curve does not seem to indicate that the difference is that large after 3 months.)

2. I think the selection bias between MV-PCI and CO-PCI is difficult to eliminate completely. However, I am most concerned about the possibility that the lesions judged to have a second culprit lesion may have differed significantly between the MV-PCI and CO-PCI groups. Please enlighten us on this point.

3. Please correct the following points.
Line 158 (P4) exclusion → occlusion
P5-6 RCX→LCX?

Reviewer 2 Report

We would like to thank the authors for the opportunity to read and review their article "Multivessel versus culprit-only percutaneous coronary intervention in patients with non-ST-elevation acute coronary syndrome". In this article, the authors showed that NSTE-ACS patients with MVD undergoing MV-PCI have similar mortality rates at long-term follow-up compared to CO-PCI. However, improved event-free survival in terms of fewer coronary reinventions was observed. This article, when reviewed, caused conflicting emotions. On the one hand, it presents data from a national register, which makes it possible to provide information about the daily practices of a diverse ethnic and socioeconomic population. Also in the article, the statistical analysis of the results is well developed, only the listing of the methods of analysis used causes respect and involuntarily inspires confidence in the results obtained. At the same time, there are fundamental remarks to the design of the article that significantly reduce the scientific significance of the data presented.

1.      In NSTE-ACS, in contrast to STE-ACS, in which the culprit lesion can be easily identified, identification of the culprit lesion is difficult. Therefore, assigning patients to the CO-PCI group may be erroneous.

2.      In this study, treatment groups (MV-PCI or CO-PCI) were determined after PCI. Therefore, patients in whom a PCI strategy was originally planned but received only one vessel PCI, because technical or anatomical factors were classified as CO-PCI, which could lead to biased results. In other words, due to the technical impossibility of performing MV-PCI, the intervention was limited to one artery. However, in this case, in reality, the comparison was made not between MV-PCI and CO-PCI, but between complete and incomplete revascularization. As is already well known, the prognosis is worse with incomplete revascularization than with complete (1), to these data the present study did not add new facts.

3.      Another important point. If it is assumed that all stenoses in multivessel disease required stenting, then in the case of initially single-vessel stenting directly for NSTE-ACS, it was required to stent other stenoses at the second stage. The authors did not consider such planned stents as endpoints, however, failure to perform stenting at the second stage, no doubt, leads to a greater number of recurrent acute coronary syndromes and emergency PCI. Therefore, an analysis is highly desirable - how many planned stentings were planned for patients with CO-PCI, how many were actually done, and what was the prognosis in these subgroups.

4.      Due to the limited data in the registry (only a small part of the basic and procedural characteristics are given), there is no data on the extent of coronary artery disease (for example, according to the SYNTAX scale), comorbid pathology (for example, kidney damage), technical possibilities for performing MV-PCI. As a result, the data obtained by the authors have low scientific significance.

References:

1.      Hambraeus K, et al. Long-Term Outcome of Incomplete Revascularization After Percutaneous Coronary Intervention in SCAAR (Swedish Coronary Angiography and Angioplasty Registry). JACC Cardiovasc Interv. 2016 Feb 8;9(3):207-215. doi: 10.1016/j.jcin.2015.10.034.

Reviewer 3 Report

Interesting article on the comparison between Culprit PCI and Multivessel PCI

Interesting article on the comparison between Culprit PCI and Multivessel PCI

Interessante articolo sul confronto tra Culprit PCI e Multivessel PCI

Interesting article about the comparison of Culprit PCI and Multivessel PCI

Articolo interessante sul confronto tra Culprit PCI e Multivessel PCI

Impossibile caricare i risultati completi

Riprova

Nuovo tentativo…

Impossibile caricare i risultati completi

Riprova

Nuovo tentativo…

Minor remarks

Minor remarks

Osservazioni minori

Minor observations

Osservazioni minori

Impossibile caricare i risultati completi

Riprova

Nuovo tentativo…

Impossibile caricare i risultati completi

Riprova

Nuovo tentativo…

- Obviously there are all the limits of a propensity matching, important numbers anyway

- Obviously there are all the limits of a propensity matching, important numbers anyway

- Ovviamente ci sono tutti i limiti di un abbinamento di propensione, numeri comunque importanti

- Obviously there are all the limits of a propensity matching, however important numbers

- Ovviamente ci sono tutti i limiti di un abbinamento di propensione, numeri comunque importanti

Impossibile caricare i risultati completi

Riprova

Nuovo tentativo…

Impossibile caricare i risultati completi

Riprova

Nuovo tentativo…

- Cite that the results of some randomized trials on the subject will soon be available (FIRE Trial)

- Cite that the results of some randomized trials on the subject will soon be available (FIRE Trial)

- Citare che i risultati di alcuni studi randomizzati sull'argomento saranno presto disponibili (FIRE Trial)

- Cite that results of some randomized trials on the subject (FIRE Trial) will soon be available

- Citare che i risultati di alcuni studi randomizzati sull'argomento (FIRE Trial) saranno presto disponibili

Impossibile caricare i risultati completi

Riprova

Nuovo tentativo…

Impossibile caricare i risultati completi

Riprova

Nuovo tentativo…

- the neutral impact on mortality has to guide the discussion

- the neutral impact on mortality has to guide the discussion

- l'impatto neutro sulla mortalità deve guidare la discussione

- the neutral impact on mortality should guide the discussion

- l'impatto neutro sulla mortalità dovrebbe guidare la discussione

Impossibile caricare i risultati completi

Riprova

Nuovo tentativo…

Impossibile caricare i risultati completi

Riprova

Nuovo tentativo…

- Revascularization remains a weak endpoint

- Revascularization remains a weak endpoint

...

- Revascularizations are still a weak endpoint

...

Impossibile caricare i risultati completi

Riprova

Nuovo tentativo…

Impossibile caricare i risultati completi

Riprova

Nuovo tentativo…

- Strange that the results are not affected by age

- Strange that the results are not affected by age

- Strano che i risultati non siano influenzati dall'età

- Strange that the results are not influenced by age

- Strano che i risultati non siano influenzati dall'età

Impossibile caricare i risultati completi

Riprova

Nuovo tentativo…

Impossibile caricare i risultati completi

Riprova

Nuovo tentativo…

- No significant variables emerge except perhaps diabetes

- No significant variables emerge except perhaps diabetes

- Non emergono variabili significative tranne forse il diabete

- No significant variables emerge, apart from diabetes perhaps

- Non emergono variabili significative, a parte forse il diabete

Impossibile caricare i risultati completi

Riprova

Nuovo tentativo…

Impossibile caricare i risultati completi

Riprova

Nuovo tentativo…

- How do you explain the role in favor of MV-PCI in women?

Round 2

Reviewer 2 Report

I carefully read the responses of the authors to the comments. The design of the manuscript, due to initial limitations (data obtained from a national register with a small set of indicators), cannot be improved. The answers of the authors testify to the same. I think that part of the reasoning from the authors' response deserves to be included in the text of the manuscript. The possibility of the manuscript publication in this design is at the discretion of the editor, my opinion remains the same.
